# OpenReview forum: "DeFacto: Counterfactual Thinking with Images for Enforcing Evidence-Grounded and Faithful Reasoning"
_ICML.cc/2026/Conference — ICML 2026 regular_

### Official Review · Reviewer_MHQo · 2026-02-20

**Soundness:** 3
**Presentation:** 2
**Significance:** 3
**Originality:** 3
**Overall Recommendation:** 4
**Confidence:** 4

**Summary:**

This paper proposes DeFacto, a counterfactual thinking-with-images framework designed to enforce evidence–answer consistency in MLLMs. The authors identify three common failure modes in visual reasoning, including mislocalized failure, spurious correctness, and faithful incorrectness—and argue that existing methods optimize answer accuracy without explicitly ensuring alignment between selected visual evidence and final predictions. To address this, they introduce a training approach that encompasses positive, counterfactual, and random-masking instances, optimized with GRPO-based reinforcement learning. They also design a composite reward that jointly optimizes answer correctness, format consistency, and region-selection coherence. To support the training, they construct DeFacto-100K, an automatically generated counterfactual dataset, and DeFacto-1.5K, a human-annotated benchmark for evaluating region-level faithfulness using detection-style metrics. Extensive experiments across general VQA, document VQA, and text-centric benchmarks show that DeFacto improves both answer accuracy and evidence-grounded consistency over strong baselines.

**Compliance With Llm Reviewing Policy:**

Affirmed.

**Key Questions For Authors:**

Table 2 shows noticeable accuracy drops on DocVQA (−5.2%) and TextVQA (−3.2%). Can you suggest the potential reasons?

**Limitations:**

No, but I don't see any negative societal impact about this work.

**Strengths And Weaknesses:**

Strength:

1. The proposed automatic counterfactual data construction method is scalable and practical. Leveraging Qwen2.5-VL for descriptor extraction and DINO-X for grounding provides a reasonable automatic proxy for evidence localization without human annotation.
2. The proposed DeFacto-1.5K benchmark with human-annotated evidence regions is a significant contribution to the evaluation of multimodal reasoning. Evaluating the faithfulness of grounding with detection-style metrics (mAP/AP50/AP75 + IoU) in conjunction with answer accuracy provides a more rigorous evaluation of faithful thinking-with-image reasoning than accuracy alone.
3. Broad evaluation and strong empirical validation. The experiments span a wide range of benchmarks, including general VQA, document understanding, and scene text reasoning tasks. The proposed method consistently improves evidence–answer consistency and grounding faithfulness over strong baselines, lending credibility and robustness to the empirical claims.

Weakness:

1. The paper suffers from several formatting and clarity problems that hinder readability. For example, Lines 241–259 contain large blank spaces and disrupted paragraph structure; Equation (6) exceeds the right margin; and the conclusion is placed in the appendix rather than in the main body, which is unconventional. The long list of benchmarks in Lines 241–259 would be better moved to the appendix for improved flow.
2. Limited analysis of accuracy degradation on some benchmarks. While the method improves faithfulness, Table 2 shows noticeable accuracy drops on DocVQA (−5.2%) and TextVQA (−3.2%). The paper lacks discussion or hypotheses explaining these degradations. A deeper analysis of why counterfactual training may hurt performance in certain text-centric or document reasoning scenarios would strengthen the empirical insights.
3. Evaluation restricted to a single backbone model. All experiments are conducted solely on the Qwen2.5-VL backbone. This makes it difficult to assess whether the proposed framework generalizes across different MLLMs. Applying DeFacto to at least one additional base model on a subset of benchmarks would significantly strengthen the claim of broad applicability.

---

> ### Author Rebuttal · Authors · 2026-03-30
>
> **Response to Weakness 1:**
> We thank the reviewer for this helpful suggestion. In the revised version, we have fixed the formatting issues and improved the paragraph structure. Specifically, the long benchmark list in Lines 241–259 has been moved to **Appendix B** for better readability, and the Conclusion has been moved back to the main paper. We have also reformatted Eq. (6) and Eq. (8) for improved clarity.
>
> **Response to Weakness 2:**
> We thank the reviewer for this helpful suggestion. We agree that a deeper analysis is needed for the accuracy drop on DocVQA and TextVQA. A main reason is that these tasks are highly **fine-grained and text-sensitive**: the model must not only identify the relevant region, but also preserve very small textual details after resizing and masking.
>
> For example, in **Appendix C, Figure 18**, the question asks: *“What company had a share of 16.5 percent of the world liner fleet?”* To answer this correctly, the model needs to simultaneously locate the tiny numeric cue **“16.5%”** and align it with the correct row label **“Mediterranean Shg Co”** in the bar chart. Both the number and the corresponding text are visually small and spatially separated, making the reasoning especially sensitive to image resizing and region masking. Similar issues also appear in Figures 6 and 17, where the key evidence consists of small textual regions or fine document details.
>
> For such extremely fine-grained textual details, the model becomes sensitive to small visual distortions (e.g., after image resizing), which can lead to errors. We plan to develop methods specifically for fine-grained text reasoning to better handle these cases in future work.
>
> **Response to Weakness 3:**
> We thank the reviewer for this helpful suggestion. To further examine generalization across backbones, we additionally validate DeFacto on **InternVL3.5-2B**. The results below show consistent improvements from the base model to SFT, GRPO, and our full DeFacto design.
>
> | Dataset | InternVL3.5-2B | + SFT | + GRPO | + DeFacto (Full) |
> |--------|---------------:|------:|-------:|-----------------:|
> | AI2D     | 64.70  | 65.51  | 66.98  | **69.93** |
> | ChartQA  | 37.76  | 39.16  | 39.76  | **40.96** |
> | OKVQA    | 44.43  | **45.78** | 45.38  | 45.60  |
> | TextVQA  | 70.66  | 72.02  | 73.18  | **74.42** |
> | VizWiz   | 23.69  | 24.29  | 24.56  | **25.60** |
>
> These results suggest that the effectiveness of DeFacto is not limited to Qwen2.5-VL and can transfer to other MLLM backbones as well. We will include this analysis in the revised paper.

---

> > ### Author Rebuttal · Reviewer_MHQo · 2026-04-03
> >
> > The authors added experiments and promised to improve the writing quality to address my concerns, however, I need to consider the quality of the original submission and retain my score of weak accept.

---

> > > ### Author Response · Authors · 2026-04-04
> > >
> > > Thank you for your careful consideration and for acknowledging our efforts. We sincerely appreciate your recognition of our work. We will incorporate all the additional experiments and writing improvements into the revised version to further strengthen the original submission.

---

### Official Review · Reviewer_T67c · 2026-03-05

**Soundness:** 2
**Presentation:** 3
**Significance:** 2
**Originality:** 2
**Overall Recommendation:** 4
**Confidence:** 2

**Summary:**

This paper introduces DeFacto framework to improve the truthfulness and grounding of multimodal large language models during visual reasoning by leveraging counterfactual training. The authors also constructed DeFacto-100K from the existing VQA datasets and annotated DeFacto-1.5K.

**Compliance With Llm Reviewing Policy:**

Affirmed.

**Final Justification:**

My main concern in the original submission was whether the evidence was strong enough to support the claim of jointly improving answer correctness and region-level faithfulness. The authors have addressed this concern by adding further experiments and analyses targeting the failure modes in Figure 1. These additions have increased my confidence in the work, and I will recommend a weak accept for this paper.

**Key Questions For Authors:**

1. How would the authors justify evaluating primarily answer accuracy given their claimed contribution of optimizing region-level faithfulness?
2. Can the authors explain why the gains in Tables 1 and 2 do not translate as strongly to the external benchmarks reported in Table 5?
3. Could the authors clarify the human verification process for evidence annotations, such as the annotator instructions, the verification criteria, how disagreements were resolved, and any inter-annotator agreement statistics?

**Limitations:**

Yes.

**Strengths And Weaknesses:**

Soundness: In my opinion, the submission is not technically sound for the following reasons.
First, the claims are not well supported. Specifically, the authors claim that “DeFacto jointly optimizes answer correctness and region-level faithfulness.” However, the main results in Tables 1 and 2 are evaluated primarily on answer accuracy, which, in my view, can support improvements in answer correctness but does not directly substantiate improvements in region-level faithfulness. The authors do present a faithfulness comparison in Figure 3, but since DeFacto-1.5K follows the same output format as the proposed training setup, this comparison may be unfair to GRIT and DeepEyes, which were not trained or designed for this specific dataset format. Second, I find the framework’s reported performance unconvincing. Although DeFacto appears to outperform baselines in Tables 1 and 2 across the VQA datasets in answer accuracy, 13 of these 15 datasets were also used to construct the DeFacto-100K training dataset. This raises the concern that the gains may partly reflect dataset-specific tuning rather than robust generalization. In Table 5,  the authors report additional results on other benchmarks, where DeFacto appears close to or in some cases worse than zero-shot GPT-4o on 3 out of 4 benchmarks, and it is also outperformed by Visual-SR1 and Chain-of-Focus on MMMU and POPE.

Presentation: The paper is clearly written, but I think the structure could be improved. In particular, I believe Table 5 and the Conclusion are important enough to be included in the main paper rather than placed in the appendix, as they currently are.

Significance: The authors attpempted improving faithful reasoning in thinking-with-images, which is an important problem compare to the exising benchmarks. But as mentioned earlier, the experimental resutls are not so convencing to me especially those results in Table 5. Also, I believe the provided human verfication benchmark would be valuable and appreciated, although the annotation size is relatively small, which may limit its practical impact and the strength of the conclusions drawn from it.

Originality: I think their attempt to improve unfaithful reasoning would advance thinking-with-images research, but again there are many concerns and drawbacks in the work as mentioned earlier.

---

> ### Author Rebuttal · Authors · 2026-03-28
>
> **Response to Weakness 1 Faithfulness Evaluation and Fairness :**
> We thank the reviewer for this helpful suggestion. Tables 1 and 2 and Figure 3 are designed as **parallel evaluations**, rather than sequential ones: Tables 1 and 2 measure answer correctness, while Figure 3 evaluates region-level faithfulness using grounding metrics such as IoU and mAP. Thus, our method is assessed simultaneously along both dimensions.
>
> Regarding fairness, when evaluating DeepEyes, GRIT, and other baselines, we strictly follow their original prompts and inference procedures. Our dataset is only used as ground-truth for evaluation, without modifying their outputs or formats. For example, we compare DeepEyes’ CoT-based crop regions and GRIT’s predicted coordinates with ground-truth boxes using IoU/mAP. Therefore, the comparison does not favor our method but provides a unified evaluation of faithfulness across models.
>
>
> **Response to Weakness 2 Generalization Beyond Training Datasets:**
> We thank the reviewer for this helpful suggestion. To address the concern of dataset-specific tuning, We have added evaluations on external benchmarks including V*, HRBench4K, HRBench8K, MathVista, MathVision, and TreeVGR, which are not used in constructing DeFacto-100K. Due to space limitations, we do not reproduce all tables here. Results on V* and related benchmarks can be found in **Reviewer pqQq Table 1**, while the full **TreeVGR results** are available at:
>
> https://anonymous.4open.science/r/icml_rebuttal_defacto-B0ED/treevgr_results.md. In addition, the ablation study in Table 3 demonstrates that our full reward design consistently outperforms SFT and standard GRPO under the same data, further supporting that the gains come from the method itself rather than dataset bias.
>
> **Response to Weakness 3 External Benchmark Competitiveness:**
> We thank the reviewer for this helpful suggestion. GPT-4o is a much larger model than ours, and our method remains competitive with it despite the substantial model scale gap.
>
> While some individual metrics are surpassed by Visual-SR1, our method achieves the best overall performance among thinking-with-images approaches. In particular, on POPE, our result is **not lower** than Chain-of-Focus but slightly higher (**88.6 vs. 88.4**). This shows that DeFacto remains highly competitive even when compared with recent strong grounded-reasoning methods.
>
> | Model | OCRB | MMstar | MMMU | MMB1.1 | POPE | Avg |
> |------|-----:|-------:|-----:|-------:|-----:|----:|
> | GRIT | 322 | 36.3 | 17.1 | 9.7 | 85.7 | 43.9 |
> | ViCrop | 233 | 33.1 | 26.1 | 51.7 | 87.3 | 66.7 |
> | DeepEyes | 636 | 43.6 | 44.1 | 29.4 | 87.7 | 62.2 |
> | Visual-SR1 | 449 | 62.8 | 57.2 | 77.4 | 86.0 | 82.2 |
> | Chain-of-Focus | 632 | 58.1 | 46.1 | 75.3 | 88.4 | 81.9 |
> | Pixel Reasoner | 597 | 62.9 | 52.5 | 78.5 | 87.8 | 83.2 |
> | DeFacto (Ours) | 871 | 63.2 | 56.6 | 81.2 | 88.6 | **85.3** |
>
> **Response to Presentation:**
> We thank the reviewer for this helpful suggestion. We will move Table 5 and the Conclusion into the main paper in the revised version.
>
>
> **Response to Significance:**
> We thank the reviewer for this helpful suggestion. In future work, we plan to develop more automated annotation methods for this type of data and further scale up the dataset.
>
> **Response to Key Questions 1:**
> We do not treat answer accuracy as the primary metric; instead, answer correctness and region-level faithfulness are evaluated in parallel. We include both accuracy (Tables 1 and 2) and grounding-based metrics (e.g., IoU and mAP) to jointly assess performance, while following the original settings of prior methods and using our annotations only as ground truth.  In addition, we have added evaluations on V*, HRBench4K, HRBench8K, MathVista, MathVision, and TreeVGR to further validate both generalization and grounding faithfulness.
>
> **Response to Key Questions 2**
> A main reason is that Table 5 is substantially more challenging: it includes additional external public benchmarks and compares against several much larger models, such as GPT-4o and LLaMA-3.2-90B-Vision. Even so, as shown in Table 5, our method remains strongly competitive. In addition, the ablation results in Table 3 further support the effectiveness of the full design: the full reward version consistently outperforms SFT, SFT with our data, and the standard GRPO variant.
>
> **Response to Key Questions 3:**
> We thank the reviewer for this helpful question. Given a question and image, annotators select one or more bounding boxes most relevant to answering the question.
>
> The process includes two stages: (1) Qwen2.5-VL-32B generates initial annotations; (2) human annotators verify and correct them. We identified around **114 problematic samples**, which were manually corrected.
>
> Some visualization examples are available at:
> https://anonymous.4open.science/r/icml_rebuttal_defacto-B0ED/example1.png to
> https://anonymous.4open.science/r/icml_rebuttal_defacto-B0ED/example7.png

---

> > ### Author Rebuttal · Reviewer_T67c · 2026-04-03
> >
> > Thank you for the detailed rebuttal. I appreciate the additional experiments and clarifications. However, my main reservations are not fully resolved. In particular, I remain unconvinced that the evidence provided is sufficient to strongly support the claim of jointly improving region-level faithfulness, as illustrated in Figure 1, since this is precisely the key limitation the paper identifies in existing methods such as DeepEyes and GRIT.
> >
> > I appreciate the additional experiments and analyses, and I hope they will be included in the revised version. These results help address my concerns. I will raise my score to weak accept.

---

> > > ### Author Response · Authors · 2026-04-04
> > >
> > > **Response to Concern on Region-level Faithfulness**
> > >
> > > We thank the reviewer for the thoughtful follow-up. To directly address the concern that the current evidence may still be insufficient to strongly support our claim of jointly improving answer correctness and region-level faithfulness, we conduct **five additional analyses on the DeFacto-1.5K benchmark**.
> > >
> > > In particular, these experiments are designed to explicitly target the three failure modes illustrated in Figure 1—namely **mislocalized**(incorrect regions leading to wrong answers), **spurious correctness** (correct answers without proper evidence grounding), and **faithful incorrectness** (correct regions but wrong answers).
> > > Specifically, we evaluate:
> > >
> > > 1. **Mislocalized & Wrong**: the predicted region does not match the annotated evidence region, and the final answer is also incorrect. This measures failures caused by incorrect grounding.
> > > 2. **Spurious Correct**: the predicted region is incorrect, but the final answer is correct. This captures shortcut-based or hallucinated success without proper evidence grounding.
> > > 3. **Faithful but Wrong**: the predicted region matches the annotated evidence region, but the final answer is still incorrect. This separates reasoning errors from grounding errors.
> > > 4. **Masked Evidence Abstention**: we mask the human-annotated key evidence regions in the original image, and test whether the model correctly abstains (i.e., predicts *Unknown* rather than guessing). This directly measures whether the model truly depends on the required evidence.
> > > 5. **Image Replacement Abstention**: we replace the original image with another image from the same dataset while keeping the question unchanged, and again test whether the model correctly abstains. This evaluates robustness against irrelevant visual input.
> > >
> > > Averaged over all 15 datasets in DeFacto-1.5K, the results are:
> > >
> > > | Metric | DeepEyes | GRIT | DeFacto (Ours) |
> > > |---|---:|---:|---:|
> > > | Mislocalized & Wrong ↓ | 55.4 | 67.9 | **26.1** |
> > > | Spurious Correct ↓ | 19.5 | 8.4 | **5.6** |
> > > | Faithful but Wrong ↓ | 6.3 | 7.6 | **4.3** |
> > > | Masked Evidence Abstention Acc ↑ | 39.1 | 46.1 | **64.1** |
> > > | Image Replacement Abstention Acc ↑ | 40.3 | 46.3 | **61.4** |
> > >
> > > Compared with DeepEyes and GRIT, DeFacto substantially reduces mis-grounded failures and spurious correct predictions, while achieving much stronger abstention behavior when the required evidence is removed or replaced. This provides stronger evidence that our gains are not only from answer accuracy, but also from genuinely more faithful evidence-grounded reasoning.
> > >
> > > Due to space limitations, the full per-dataset results are available at:
> > > https://anonymous.4open.science/r/icml_rebuttal_defacto-B0ED/new_results.md

---

### Official Review · Reviewer_cWe5 · 2026-03-10

**Soundness:** 3
**Presentation:** 3
**Significance:** 2
**Originality:** 2
**Overall Recommendation:** 4
**Confidence:** 3

**Summary:**

This paper proposes DeFacto, a framework for improving evidence-grounded reasoning in multimodal large language models (MLLMs). The authors observe that existing “thinking-with-images” approaches often produce answers that are inconsistent with the visual evidence used during reasoning, leading to unfaithful predictions. DeFacto addresses this issue through a counterfactual training paradigm combining three supervision settings: positive examples with full evidence, counterfactual examples where evidence regions are masked (requiring abstention), and random masking of irrelevant regions. The model is optimized with GRPO-based reinforcement learning using rewards for answer correctness, structured reasoning outputs, and region–answer alignment. The authors also introduce DeFacto-100K, a counterfactual training dataset, and DeFacto-1.5K, a human-annotated benchmark for evaluating evidence–answer consistency. Experiments show improvements in both answer accuracy and evidence-grounded reasoning across multiple visual question answering benchmarks.

**Compliance With Llm Reviewing Policy:**

Affirmed.

**Final Justification:**

From the responses from the authors during the rebuttal, most of my concerns are addressed, and improved performance on diverse range of benchmark shows that the proposed method is effective. Therefore, I will raise my score to weak accept.

**Key Questions For Authors:**

- How does DeFacto differ conceptually and empirically from prior counterfactual/causal VQA work [3], rather than only from recent region-grounding RL methods?

[3] Eyes Can Deceive: Benchmarking Counterfactual Reasoning Capabilities of Multimodal Large Language Models

**Strengths And Weaknesses:**

### Strengths
- Address a fundamental problem in MLLMs’ reasoning, where models rely on language priors rather than visual evidence that require fine-grained perception.
- The introduction of DeFacto-100K and DeFacto-1.5K provides useful resources for studying evidence-grounded reasoning.
- This paper propose a clear reward design for RL training to achieve the evidence-grounded reasoning.

---

### Weaknesses
* **Limited related-work coverage.**
    * The related-work discussion is incomplete, particularly regarding prior work addressing fine-grained visual perception and reliable multimodal reasoning. While the paper covers several recent “thinking-with-images” and grounding-based methods, it omits or under-discusses relevant research directions such as multimodal verification, traceable evidence evaluation, and counterfactual/causal VQA.
        * For example, the paper frames its contribution as improving evidence–answer consistency and faithful multimodal reasoning, yet it does not discuss verification-oriented approaches such as MM-Verify [1], which also aims to improve multimodal reasoning reliability by verifying reasoning trajectories. Even though MM-Verify is not explicitly region-grounding-based, it addresses a similar failure mode of producing plausible but visually not grounded reasoning.
        * In addition, the paper does not compare with recent work such as PAPO [2], which improves multimodal reasoning by jointly learning perception and reasoning through implicit perception losses to enhance visual grounding. Including comparisons or clearer discussion of differences would help better position the contribution.
* **Limited evaluation of grounded / traceable evidence.**
    * Although the paper introduces the DeFacto-1.5K benchmark to evaluate evidence–answer consistency, the evaluation remains relatively limited. To more rigorously verify the effectiveness of the proposed region reward mechanism, it would be beneficial to include additional benchmarks such as TreeVGR [4], which explicitly evaluates traceable evidence, bounding-box grounding quality, and joint localization–reasoning performance.
* **Limited comparison with existing “thinking-with-images” methods.**
    * The experimental comparison with recent “thinking-with-images” approaches is relatively limited. While the paper includes GRIT as a representative baseline, it does not evaluate against other relevant methods such as Visual-RFT or SLVR, which also aim to improve grounded visual reasoning in multimodal models. Including these baselines would provide a more comprehensive evaluation and help better assess the relative advantages of the proposed approach.
* **Insufficient ablation analysis of reward components.**
    * The method introduces multiple reward components (answer correctness, format consistency, and region selection coherence), but the paper does not fully analyze the contribution of each component. More detailed ablation studies would strengthen the claims. For example:
        * How does each reward term individually affect performance?
        * How well does the model predict bounding boxes under different reward settings?
        * What is the relationship between bounding-box localization accuracy and final answer accuracy?
    * Such analysis would provide clearer insight into why the method works.
* **Unclear accuracy of the automated evidence extraction pipeline.**
    * The dataset construction relies on an automated pipeline that extracts evidence regions using descriptor extraction and open-vocabulary detection. However, the paper does not evaluate the quality of this pipeline.
        * How is the threshold in Eq. (3) chosen for determining evidence regions?
        * What are the precision and recall of the automatically extracted bounding boxes relative to ground-truth descriptors?
        * How sensitive is the training process to errors in the extracted evidence regions?
* **Equation should be revised for better readability (equation 6&8)**

[1] MM-Verify: Enhancing Multimodal Reasoning with Chain-of-Thought Verification \
[2] Perception-Aware Policy Optimization for Multimodal Reasoning \
[3] Eyes Can Deceive: Benchmarking Counterfactual Reasoning Capabilities of Multimodal Large Language Models \
[4] Traceable Evidence Enhanced Visual Grounded Reasoning: Evaluation and Methodology

---

> ### Author Rebuttal · Authors · 2026-03-30
>
> **Response to Weakness: 1**
> Thank you for pointing out these relevant directions and references. We have expanded the related-work discussion and will include the following recent works in the revised paper, including verification-based methods such as MM-Verify and representative counterfactual reasoning works:
>
> 1. Eyes Can Deceive: Benchmarking Counterfactual Reasoning Capabilities of Multimodal Large Language Models
> 2. Counterfactual contrastive learning for weakly-supervised vision-language grounding
> 3. What if the tv was off? examining counterfactual reasoning abilities of multi-modal language models
> 4. Counterfactual vision and language learning
> 5. Learning chain of counterfactual thought for bias-robust vision-language reasoning
> 6. CounterCurate: Enhancing physical and semantic visio-linguistic compositional reasoning via counterfactual examples
> 7. Counterfactual-based saliency map: Towards visual contrastive explanations for neural networks
> 8. Dig-in: Diffusion guidance for investigating networks—uncovering classifier differences, neuron visualisations and visual counterfactual explanations
> 9. Counterfactual VQA: A cause-effect look at language bias
>
> We also thank the reviewer for suggesting **PAPO** as an additional comparison.
>
> | Model | VQAv2 | OKVQA | SciQA | VSR | DVQA | TextVQA | Avg |
> |-------|------:|------:|------:|----:|-----:|--------:|----:|
> | PAPO  | 69.9  | **58.8**  | 83.4  | 62.1 | **88.1** | 70.9    | 72.2  |
> | Ours  | **73.8** | 56.3 | **92.3** | **77.0** | **86.8** | **75.9** | **77.0** |
>
> We will include this comparison and the corresponding discussion in the revised paper.
>
> **Response to Weakness: 2**
> We thank the reviewer for this helpful suggestion. Since the full TreeVGR comparison table is too large to include here, we provide the complete results at the anonymous link below:
>
> https://anonymous.4open.science/r/icml_rebuttal_defacto-B0ED/treevgr_results.md
>
> We show a compact comparison with several key baselines here:
>
> | Model | Overall | mIoU | Attributes | OCR | Comparison |
> |---|---:|---:|---:|---:|---:|
> | Qwen2.5-VL-7B | 37.0 | – | 55.2 | 27.9 | 43.2 |
> | DeepEyes-7B | 37.5 | 30.0 | 62.1 | 51.5 | 51.7 |
> | TreeVGR-7B | 50.4 | 44.0 | 65.5 | 63.3 | 69.0 |
> | DeFacto-7B | 41.7 | 36.4 | 58.6 | 41.2 | 45.5 |
> | Δ vs Qwen2.5-VL-7B | +4.7 | – | +3.4 | +13.3 | +2.3 |
>
> Although our performance is lower than TreeVGR on some metrics, we emphasize that our model is evaluated in a **zero-shot setting without training on the TreeVGR dataset**. Meanwhile, compared with other zero-shot methods (e.g., Qwen2.5-VL-7B and DeepEyes-7B), our method shows consistent improvements and remains competitive.
>
>
> **Response to Weakness 3:**
>
> We thank the reviewer for this helpful suggestion. We have added a comparison with **Visual-RFT**. Due to space limitations, we provide the full results at:
> https://anonymous.4open.science/r/icml_rebuttal_defacto-B0ED/defacto_vs_visualrft.md
>
> Here we report key results (e.g., average performance):
>
> | Model | Avg |
> |------|----:|
> | Visual-RFT | 44.9 |
> | **DeFacto (Ours)** | **60.8** |
>
> These results show that DeFacto achieves clearly stronger overall performance compared to recent thinking-with-images methods.
>
> **Response to Weakness 4**
> We thank the reviewer for this helpful suggestion. Due to space limitations, we refer readers to **Reviewer pqQq Tables 3 and 4** for detailed ablation results. We evaluate reward design from two aspects:
> (1) answer accuracy under different $\beta_{\text{pos}}$ and $\beta_{\text{neg}}$ settings;
> (2) bounding-box quality measured by mAP, AP50, AP75, and IoU.
>
> These results show that different settings can be competitive on individual metrics, while the current choice $(\beta_{\text{pos}}=1.0,\beta_{\text{neg}}=0.6)$ achieves the best overall balance.
>
>
> **Response to Weakness 5**
> We thank the reviewer for this helpful question. The threshold in Eq. (3) is set to **0.75** in our implementation.
>
> To evaluate the quality of the automated pipeline, we conducted manual inspection. We found **5,137 incorrect samples (≈5.1%)**, including **2,914 mis-grounded regions** and **2,223 hallucinated descriptors**. This level of noise is within an acceptable range, and the consistent performance improvements indicate that the training process is robust to such errors.
>
> **Response to Weakness 5**
> We have revised Equations (6) and (8) for improved readability.
>
> **Response to Key Question1**
> Prior work mainly formulates the task as counterfactual QA, focusing on how answers change under input perturbations (e.g., masking regions). In contrast, **DeFacto is designed for region-level faithfulness**, where the model is required not only to answer correctly but also to identify and rely on the minimal and sufficient visual evidence. This shifts the task from *causal effect analysis* to *evidence-grounded reasoning*. We will discuss this work in the revised version.

---

> > ### Author Rebuttal · Reviewer_cWe5 · 2026-04-03
> >
> > Thank you for providing detailed responses. Most of my concerns are addressed, and experimental results look promising, so I will raise my score to weak accept.

---

> > > ### Author Response · Authors · 2026-04-04
> > >
> > > We thank the reviewer for the positive feedback and for raising the score. We will incorporate the additional experiments and expand the discussion of related work in the revised version.

---

### Official Review · Reviewer_pqQq · 2026-03-13

**Soundness:** 2
**Presentation:** 2
**Significance:** 2
**Originality:** 3
**Overall Recommendation:** 4
**Confidence:** 4

**Summary:**

Existing thinking with image methods fails to ensure evidence-answer consistency. The authors propose a DeFacto model for counterfactual reasoning and contribute the DeFacto-100K and DeFacto-1.5K datasets.

**Compliance With Llm Reviewing Policy:**

Affirmed.

**Final Justification:**

From the responses from the authors during the rebuttal, most of my concerns are addressed, and improved performance on diverse range of benchmark shows that the proposed method is effective. Therefore, I will raise my score to weak accept.

**Key Questions For Authors:**

The innovation of the paper is great, focusing on counterfactual questions in MLLM. If the weaknesses are addressed, I would consider increasing my score.

**Limitations:**

Look at the weaknesses section.

**Strengths And Weaknesses:**

**Strengths**

(1)The study of counterfactual questions in MLLM is of great research significance. The consistency between evidence and answers can be ensured through (i) positive, (ii) counterfactual, and (iii) random-masking approaches, which help align consistency.


**Weaknesses**

(1)In Counterfactual Dataset Construction, how can the use of MLLM for Descriptor extraction control the extraction of key and effective information regarding objects, attributes, or relationships, and how can it ensure that this information is free from hallucinations?

(2)Thinking with images can extract visual cues from large-scale images, but lacks datasets with fine-grained perception, such as the results from V*, HRBench4K, and HRBench8K. Additionally, as a general VQA model, there seems to be no related mathematical benchmarks, such as MathVista and MathVision.

(3) There are few comparison models, and the results lack persuasiveness, such as GPT and Gemini.

(4)In the design of the reward function, the experiment lacks parameter ablation, such as the values of $\beta_{\text{pos}}$ and $\beta_{\text{neg}}$.

---

> ### Author Rebuttal · Authors · 2026-03-28
>
> **Response to Weakness 1:**
> We appreciate this important concern. We use Qwen2.5-VL as a high-level controller to extract keywords from the image with a constrained prompt, and then use these keywords as queries for DINO-X detection. The prompt explicitly asks the model to output only **visually grounded, answer-relevant noun phrases or attribute phrases**, while avoiding abstract inference, uncertain relations, or background details. For example, the prompt requires the model to:
> (1) focus only on objects/attributes directly useful for answering the question;
> (2) return short phrases such as *“red traffic light”*, *“bottle cap”*, or *“printed number 16.5”* rather than full sentences;
> (3) avoid hallucinated descriptions like invisible properties, commonsense guesses, or relations not directly supported by the image.
> The exact prompt will be included in the appendix.
>
> For OCR cases, we match the recognized text with the extracted phrases; for DINO-X cases, we directly obtain target bounding boxes. We set the threshold in Eq. (3) to **0.75**, so only high-confidence regions are kept.
> We further performed manual inspection and found **5,137 incorrect samples (≈5.1%)**, including **2,914 mis-grounded regions** and **2,223 hallucinated descriptors**. This level of noise is acceptable, and the overall performance gains confirm its limited impact.
>
> **Response to Weakness 2:**
> Thank you for your suggestion. We have added results on V*, HRBench4K, HRBench8K, MathVista, and MathVision, as shown below.
>
> #### Table 1
> | Model            | Param Size | V* Bench     | HRBench4K    | HRBench8K    | MathVista    | MathVision    |
> |------------------|------------|-------------:|-------------:|-------------:|-------------:|--------------:|
> | LLaVA-OV         | 7B         | 75.4         | 63.0         | 59.8         | 58.6         | 18.3          |
> | Qwen2.5-VL       | 7B         | 78.5         | 71.6         | 67.9         | 68.3         | 25.6          |
> | Qwen2.5-VL       | 32B        | 80.6         | 74.1         | 69.9         | –            | –             |
> | InternVL3        | 8B         | 81.2         | 70.0         | 69.3         | _71.6_       | **29.3**        |
> | Pixel-Reasoner   | 7B         | 84.3         | 74.0         | 66.9         | –            | –             |
> | DeepEyes         | 7B         | **85.6**      | 75.1         | 72.6         | 70.1         | 26.6          |
> | Thyme            | 7B         | 82.2         | 77.0         | 72.0         | 70.0         | 27.6          |
> | MM-Eureka        | 7B         | –            | –            | –            | _72.6_       | 28.1          |
> | ThinkLite        | 7B         | –            | –            | –            | 71.6         | 24.6          |
> | VL-Rethinker     | 7B         | –            | –            | –            | **73.7**     | 28.4          |
> | VLAA-Thinker     | 7B         | –            | –            | –            | 71.7         | 24.2          |
> | DeepEyesV2       | 7B         | 81.8         | _77.9_       | _73.8_       | 71.9         | _28.9_        |
> | **Ours**         | 7B         | 85.3    | **78.9**     | **76.3**     | 72.2         | **29.3**      |
>
> Our method achieves competitive performance across these benchmarks. We will include these results in the revised paper.
>
>
> **Response to Weakness 3:**
> We provide a direct comparison with GPT-4o, GPT-4V, and Gemini in the table below, where our method remains competitive. More comprehensive results can be found in Appendix A (Table 5), where we evaluate on 5  benchmarks and compare with 21 open-source and closed-source models.
> #### Table 2
> | Model            | OCRB | MMstar | MMMU | MMB₁.₁ | POPE |
> |------------------|-----:|-------:|-----:|-------:|-----:|
> | GPT-4o           | 736  | **63.9** | **69.2** | **82.2** | –   |
> | GPT-4V           | 656  | 56.0  | 61.7 | 79.8  | –   |
> | Gemini-1.5-Pro   | 754  | –     | 62.2 | –     | –   |
> | **Ours**         | **871** | 63.2  | 56.6 | 81.2  | **88.6** |
>
> **Response to Weakness 4:**
> We provide ablation results on different choices of $\beta_{\text{pos}}$ and $\beta_{\text{neg}}$ as shown below. The current setting ($\beta_{\text{pos}}=1.0$, $\beta_{\text{neg}}=0.6$) achieves the best overall performance. We will include this analysis in the revised paper.
>
> #### Table 3
> | β_pos | β_neg | VQAv2 | OKVQA | SciQA | VSR  | DVQA | TextVQA | Avg  |
> |------:|------:|------:|------:|------:|-----:|-----:|--------:|-----:|
> | 0.4   | 0.4   | 70.8  | 53.4  | 89.3  | 74.1 | 83.9 | 73.0    | 74.1 |
> | 1.0   | 0.4   | 73.5  | **56.4** | 91.9  | 76.6 | 86.2 | 75.4    | 76.7 |
> | **1.0** | **0.6** | **73.8** | 56.3  | **92.3** | **77.0** | **86.8** | **75.9** | **77.0** |
> #### Table 4
> | β_pos | β_neg | mAP | AP50 | AP75 | IoU |
> |------:|------:|----:|-----:|-----:|----:|
> | 0.4   | 0.4   | 31.8 | 46.2 | 32.1 | 45.5 |
> | 1.0   | 0.4   | 34.9 | 49.2 | 34.2 | **49.6** |
> | **1.0** | **0.6** | **35.6** | **50.4** | **35.3** | 49.2 |

---

> > ### Author Rebuttal · Reviewer_pqQq · 2026-04-03
> >
> > Thank you for providing detailed responses. Most of my concerns are addressed, and experimental results look promising, so I will raise my score to weak accept.

---

> > > ### Author Response · Authors · 2026-04-04
> > >
> > > We sincerely thank the reviewer for the positive feedback and for recognizing the value of our work. We are glad that our additional analyses and clarifications have addressed your concerns and that the experimental results are now more convincing. Your insightful comments have been very helpful in improving both the clarity and quality of the paper.

---

### Decision · Program_Chairs · 2026-04-30

**Decision:**

Accept (regular)

**Comment:**

**Paper Summary:** This paper presents DeFacto, a counterfactual reasoning framework that aims to improve the reasoning of MLLMs that optimizes for answer-evidence consistency alongside task accuracy. This paper also presents two datasets, the larger DeFacto-100K and the smaller, human annotated DeFacto-1.5K.

**Strengths/Weaknesses:** The reviews for this paper are overall positive, with all reviewers recommending acceptance. Overall, they acknowledge the importance of the problem area as overreliance on linguistic priors is a known problem with VLMs. One strength is around the benchmark artifacts, particularly in how the generation pipeline lends itself toward scalability. They also note the framework uses a clear training strategy, with 3 different approaches (positive, counterfactual and masking) for alignment. Finally, DeFacto as a framework shows demonstrable empirical improvements across different benchmarks.

Most of the core weaknesses were addressed in the rebuttal and the reviewers found them satisfactory. Remaining unresolved weaknesses are whether improvements in overall answer correctness directly equates to improvements in region-level faithfulness. The authors provided an additional finegrained experiment which appears to demonstrate region-level improvements across 5 different metrics.


**Recommendation:** I recommend this paper be accepted. For the camera ready, I suggest the authors i) include the additional models models in the main figure ii) expand the related works section (*Reviewer cWe5*) iii) reformat equations (*Reviewer cWe5, Reviewer MHQo*) and check the overall structure (*Reviewer T67c*) and iv) add the region-level faithfulness experiments to the appendix.